# Strategies to Optimize Adherence in Patients with Mycosis Fungoides

**DOI:** 10.3390/cells11010113

**Published:** 2021-12-30

**Authors:** Warren H. Chan, Daniel J. Lewis, Madeleine Duvic, Steven R. Feldman

**Affiliations:** 1Center for Dermatology Research, Department of Dermatology, Wake Forest School of Medicine, Winston-Salem, NC 27157, USA; sfeldman@wakehealth.edu; 2Department of Dermatology, University of Pennsylvania, Philadelphia, PA 19104, USA; Daniel.lewis@pennmedicine.upenn.edu; 3Department of Dermatology, MD Anderson Cancer Center, University of Texas, Houston, TX 77030, USA; mduvic@mdanderson.org; 4Department of Pathology, Wake Forest School of Medicine, Winston-Salem, NC 27104, USA

**Keywords:** adherence, cutaneous T-cell lymphoma, anchoring, framing, loss aversion, mycosis fungoides, topical steroids, psoriasis, interferon, nitrogen mustard, mechlorethamine, brentuximab vedotin, mogamulizumab, romidepsin, vorinostat, PUVA, patient adherence

## Abstract

Patient adherence to medications for common skin conditions has been extensively studied over the past two decades, and suboptimal adherence is a primary contributor to treatment failure. The impact of sub-par adherence in cutaneous T-cell lymphoma (CTCL) patients has been largely unexplored, and promoting adherence in this patient population may represent a promising area of consideration for improving treatment outcomes. We apply patient adherence strategies that have been studied in dermatology to CTCL and provide concrete examples of how these strategies can be used to improve adherence in the CTCL setting. Through the implementation of small changes in how we present and counsel about therapeutic options to our patients, we can maximize patient adherence, which has the potential to optimize therapy regimens and reduce treatment failure.

Patient adherence to medications for common skin conditions has been extensively studied over the past two decades and is a primary reason for treatment failure [1]. Having a serious condition, even acute leukemia, does not imply reliable adherence to treatment [2]. Adherence in cutaneous T-cell lymphoma (CTCL) is largely unexplored and may be an exciting target for improving treatment outcomes.

Mycosis fungoides (MF) is the most common form of CTCL; at the initial diagnosis, 70% of patients have early-stage disease (IA–IIA), for which the application of topical therapies, generally more difficult than taking oral medications, is often recommended [3]. With topical steroids alone, 63% of patients with T1 (patches or plaques covering <10% body surface area [BSA]) disease and 25% with T2 (patches or plaques covering ≥10% BSA) disease achieve a complete response [4]. The difference in response rate may be at least partly attributable to lower adherence when application is needed to a larger area. Progression at five years is 10% in patients with T1 disease, 22% for T2, 48% for T3, and 56% for T4 [5]. Therefore, might we bring the rate of progression of those with T2 disease closer to those with T1, and decrease that in all stages, by improved adherence to treatment, whether topical or systemic? Patient adherence represents an area of untapped potential for improving treatment outcomes in CTCL.

Strategies to promote adherence have been extensively studied in psoriasis and can be applied to CTCL (Table 1) [1,6]. We employ the pyramid model for improving adherence: (1) building a foundation centered on trust and accountability, (2) addressing practical issues to make treatment as easy as possible for our patients, and (3) using behavioral techniques to give our patients an extra nudge to use their medications. We build trust by showing that we care (Table 1). More frequent office visits, scheduling a follow-up visit shortly after a new treatment is started, and asking patients to call or email us to report how their medication is working can build accountability. Forming good habits related to medication use in the short term may also promote better long-term use of treatment.

Next, we should make treatment as easy as possible for the patient. Can a patient on multiple topical steroids or multiple topical agents use only one instead? It may be that the reduction in adherence from adding a topical agent is not worth the marginal benefit conferred. When used consistently, one topical agent may be enough. If multiple agents are required, written patient instructions are likely to be needed. Patients tend to prefer foams and solutions over other vehicles [7]. We are taught that ointments are more potent than foams or solutions; however, ointments have zero potency if they are not used (Table 1). Patients are more likely to continue using their medication when they see that it is working; since it may take three weeks for CTCL patients to notice the difference from their treatment, asking them to call or email us after three days and then once per week for at least three weeks may build early accountability and lead to an earlier noticeable benefit, encouraging future adherence. For a similar reason, selecting fast-acting agents is helpful. Late-stage, severe, or long-standing CTCL can be depressing to patients and reduce adherence. Local radiation therapy and total skin electron beam therapy can clear even disseminated tumors and Sézary Syndrome for at least a few months, providing hope, and potentially improving adherence to future therapy and maintenance regimens. An important step involves setting expectations, as no treatments work overnight. Incorporating social work to help patients more easily navigate the treatment process and using institutional specialty pharmacies to aid in regularly scheduled calls to assess adherence are also recommended.

Lastly, we can employ behavioral techniques to nudge our patients in the right direction. Anchoring involves presenting an idea that is less desirable first to “anchor” the patient, making the next idea more desirable in comparison. For example, if we are recommending interferon to a patient, we might say, “Interferon is like insulin; it is given by injection. You are familiar with how patients with diabetes give themselves insulin injections twice a day, right? Well, this medication is not exactly like insulin—you only need to take the medication twice per week.” Another behavioral technique involves the observation that humans tend to gravitate towards, and base their therapy decisions on, salient anecdotes more often than data. We can use saliency to our advantage by saying, “One of my other patients who reminds me of you and had disease very similar to yours had an excellent response to this medication. In fact, I think I saw that patient in this same exam room!” When counseling about adverse effects, patients tend to focus on the small risk of a side effect, rather than the comparatively large benefits of therapy. We should help reframe the patient’s perspective by stating that 99 out of 100 patients do not have a certain side effect with a medication, rather than, for example, that mechlorethamine gel carries a 1% risk of nonmelanoma skin cancer. We can also use side effects to our advantage: the skin irritation from topical retinoids is “a sign that the drug is working.”

If better adherence can contribute to a decreased progression risk, it may, for many CTCL patients, also prevent the need for systemic therapies, which are associated with high cost and side effects, which are major barriers to treatment in those with late-stage disease. Higher adherence to systemic therapies and adjuvant topical therapies may improve the patient’s response to systemic therapies. Better adherence to topical steroids may lead to better control of pruritus. We might think that because of their cancer diagnosis, CTCL patients are more likely to adhere to treatment. However, chronicity and worsened impact of disease are associated with poorer adherence [8]. CTCL, especially MF, is not likely to be an exception. We should not assume that patient adherence is the norm and should instead work even harder to promote adherence in our cancer patients, especially given the severe consequences of non-adherence.

## Figures and Tables

**Table 1 cells-11-00113-t001:** Patient adherence techniques and examples for the cutaneous T-cell lymphoma setting.

Pyramid Level	Technique	Examples
Foundation of trust and accountabiility	Showing that we care	Show up on time for clinic. Open the door to the exam room slowly to show that you are not rushed. Do not look at your watch during the patient visit.
		Wash your hands in front of the patient.
		Empathy: “I bet the previous treatments have been very frustrating, right?”
		Assess patient satisfaction via surveys
		Let the patient tell you their story without interrupting
		Make yourself accessible. One method is to give your contact info to patients
	More frequent office visits (increases white coat compliance)	Photopheresis involves two consecutive day sessions every 3–4 weeks. If possible, have patients see a CTCL dermatologist before or after each session
	Ask patients to call or email us to report how the medication is working	
Simplicity and education	Simplify the treatment regimen	Switch multiple topical steroids or multiple topical agents to one steroid or topical agent
	Easier vehicle	Switch carumustine ointment to nitrogen mustard aqueous solution or mechlorethamine gel
	Shorten initial treatment interval	Since it takes roughly 3 weeks to see improvement, ask patients to call or email you in 3 days, then once per week for at least 3 weeks
	Minimize cost of treatment	Use the EMR to auto-populate instructions for the pharmacist: “If the pharmacy offers a similar but less expensive option, feel free to switch to that medication, if the patient wants to.”
	Written instructions	Printed from EMR, tear off pads, or sticky notes: “Hydrocortisone to face, triamcinolone to body, clobetasol to palms/soles”
	Involve patients in the treatment choice	Stage IA (T1): topical steroids v. topical mechlorethamine v. topical retinoids v. phototherapy v. imiquimod v. other
		Stage IB (T2): bexarotene v. phototherapy v. interferon v. other
	Choose a fast-acting agent	Total skin electron beam therapy can clear even disseminated tumors and Sezary Syndrome for at least a few months
		Local radiation therapy is very effective at clearing discreet, even tumors
Behavioral techniques	Anchoring	“Interferon is like insulin; it is given by injection. You are familiar with how patients with diabetes give themselves insulin injections twice a day, right? Well, this medication is not exactly like insulin—you only need to take the medication twice per week.”
		Mogamulizumab: Say 4x/wk at first. Dosing actually starts weekly, then decreases to biweekly, then monthly
		Use romidepsin (3x/mo) or mogamulizumab (weekly at first) as anchors for brentuximab vedotin (once every 3 wks)
		Use vorinostat, which causes diarrhea, as an anchor for bexarotene, for which the side effects can be easily managed with pills for thyroid and cholesterol
		Use PUVA (psoralen causes diarrhea) to anchor for nb-UVB, for which no extra pill is needed
	Saliency	“One of our other patients who reminds me of you and had disease very similar to yours had an excellent response to this medication. In fact, I think I saw that patient in this same exam room.”
	Framing side effects	Nitrogen mustard has a 1–5% increased risk of developing NMSCs. Reframe as “99/100 do not have this problem.”
		There is a <0.01% chance of developing PML with BV. Reframe as “9999/10,000 do not have this problem.”
	Loss aversion (emphasize loss vs. gain)	“This drug can prevent your disease from growing worse”
	Counteracting steroid phobia	“Steroids are cortisone medications, like over-the-counter hydrocortisone, only a little stronger. All-natural, organic, gluten-free”
	Using side-effects to our advantage	Retinoids, imiquimod, resiquimod: “Skin irritation is a sign that it is working!”

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
