# Peer review of "Strategies to Optimize Adherence in Patients with Mycosis Fungoides"

_cells, 2021, doi:10.3390/cells11010113_

Round 1
Reviewer 1 Report
The purpose of this study is to discuss adherence in the treatment of CTCL.
Although articles on adherence in other dermatological illnesses have been published, there has been no equivalent paper on CTCL yet, making this work unique.
Although other reviewers may have different preferences, I suppose this paper is valuable.
I have no major concerns about the content of this paper.
However, I do have a small concern before the publication of this paper.
This research focuses on mycosis fungoides, particularly its early stage.
Indeed, mycosis fungoides, especially early-stage mycosis fungoides, accounts for more than half of all cutaneous lymphomas.
However, there are many other variants of cutaneous lymphoma besides mycosis fungoides.
I am afraid that the title "Strategies to Optimize Adherence in Patients with Cutaneous T-cell Lymphoma" is a bit exaggerated.
The phrases mycosis fungoides or early-stage mycosis fungoides should be included at the very least in the subtitle.
Reviewer 2 Report
CTCL is a chronic and slowly progressing disease. Treatment adherence is a chronic problem in early stage patients. Strategies to promote adherence have not been extensively studied and, therefore the authors propose an adherence technique that is built on a pyramid model used for psoriasis patients with steps incorporated to building trust, simplify education, and behavioral techniques. The manuscript is well written and clearly outlined and a helpful guide in managing early stage patients. perhaps a step could be implemented as to "set expectations". As none of the treatments really work over night. Incorporate social work into team that will help patients to navigate through treatment and other issues that arise (e.g access to treatment, center and medication).
Reviewer 3 Report
The authors hypothesize that less than optimal adherence to topical medications may partly account for the differences in response rates and Progression free survival durations in even T1 and T2 dz. However, despite organizations which have more than adequate patient numbers and data sets, they present no actual preliminary data to support differences in adherence in their patient populations.
In Table 1 "foundation and accountability" phototherapy is described as 2 consecutive days every 3-4 weeks. I believe they are referencing extracorporeal photopheresis, and invasive systemic therapy reserved for Sezary pts. Phototherapy is normally NB-UVB which is administered 2-3x/week at induction.
The examples in Table 1 and the text describe anchoring, but utilize again advanced systemic therapies administered often orally or intravenously. How this relates to the use of topical agents is not really clear.
No reference is made to the use of an institutional specialty pharmacy for the dispensing of the topical medications. These structured groups often provide extensive pre treatment medication counseling, and often have regular scheduled calls with patients to assess adherence. A discussion of such groups/and the possible use to improve topical adherence may be worht considering.
Round 2
Reviewer 3 Report
The authors do not present a hypothesis nor any scientific data even in pilot form to suggest the interventions work. A prospective trial or pilot study would be very helpful